# Psychoactive Drugs Induce the SOS Response and Shiga Toxin Production in *Escherichia coli*

**DOI:** 10.3390/toxins13070437

**Published:** 2021-06-23

**Authors:** John K. Crane, Mashal Salehi, Cassandra L. Alvarado

**Affiliations:** Department of Medicine, Division of Infectious Diseases, University at Buffalo, Buffalo, NY 14221, USA; salehi.mashal@yahoo.com (M.S.); clalvara@buffalo.edu (C.L.A.)

**Keywords:** serotonin selective reuptake inhibitors, phenothiazines, RecA, hypermutation, enterohemorrhagic *E. coli*, Shiga-toxigenic *E. coli*, hemolytic-uremic syndrome

## Abstract

Several classes of non-antibiotic drugs, including psychoactive drugs, proton-pump inhibitors (PPIs), non-steroidal anti-inflammatory drugs (NSAIDs), and others, appear to have strong antimicrobial properties. We considered whether psychoactive drugs induce the SOS response in *E. coli* bacteria and, consequently, induce Shiga toxins in Shiga-toxigenic *E. coli* (STEC). We measured the induction of an SOS response using a *recA-lacZ* *E. coli* reporter strain, as RecA is an early, reliable, and quantifiable marker for activation of the SOS stress response pathway. We also measured the production and release of Shiga toxin 2 (Stx2) from a classic *E. coli* O157:H7 strain, derived from a food-borne outbreak due to spinach. Some, but not all, serotonin selective reuptake inhibitors (SSRIs) and antipsychotic drugs induced an SOS response. The use of SSRIs is widespread and increasing; thus, the use of these antidepressants could account for some cases of hemolytic-uremic syndrome due to STEC and is not attributable to antibiotic administration. SSRIs could have detrimental effects on the normal intestinal microbiome in humans. In addition, as SSRIs are resistant to environmental breakdown, they could have effects on microbial communities, including aquatic ecosystems, long after they have left the human body.

## 1. Introduction

It has been recognized for several decades that psychotropic drugs can demonstrate fairly strong antimicrobial activity [1,2]. In the early years of this discovery, the antimicrobial activity was often described enthusiastically, for example, as providing new therapeutic strategies for drug-resistant pathogens. More recently, however, these antimicrobial effects of non-antibiotic drugs have been more often regarded negatively because of their ability to disrupt the gut microbiome [3]. Maier et al. reported on the effects of more than 1000 drugs commonly used in human medicine and tested them for their ability to inhibit the growth of commensal bacteria in conditions mimicking the gastrointestinal tract [4]. Drug classes that demonstrated the strongest ability to disrupt bacteria comprising the normal microbiome include proton-pump inhibitors (PPIs), non-steroidal anti-inflammatory drugs (NSAIDs), typical and atypical anti-psychotic drugs, and some antidepressant medications. 

The SOS response is a bacterial stress response pathway activated mainly by DNA damage. Among the many actions of the SOS pathway is the hypermutation response, in which the mutation rate of the bacteria is increased. We became interested in the role of SOS-inducing drugs as being possible triggers for the emergence of new antibiotic resistance. For example, exposure to ciprofloxacin triggered an increase in antibiotic resistance frequency to rifampin, chloramphenicol, and doxycycline in vitro [5]. In vivo, ciprofloxacin and the anti-retroviral drug zidovudine triggered hypermutation in the rabbit intestine, resulting in increased acquisition of new antibiotic resistance in *E. coli* [6]. We considered if, in addition to inhibiting growth of commensal microbes, as described by Maier et al. [4], psychotropic drugs also induce the SOS response. As we witnessed evidence of the induction of an SOS response in a preliminary screening assay, we next tested whether those drugs stimulated Shiga toxin production in STEC. Production of the Shiga toxins, Stx1 and Stx2, is one of the most important aspects of STEC virulence. The SOS pathway is the main regulator of Stx production in STEC [7]. Many antibiotics induce the SOS response and often trigger increased Stx toxin production by STEC, which is why the CDC and other public health agencies state that antibiotics are contra-indicated in STEC infection. 

In this study, we tested whether antidepressant medications in the serotonin selective reuptake inhibitor (SSRI) class could activate the SOS response using a *recA-lacZ* reporter strain as an initial screen. Drugs that were capable of inducing RecA expression in vitro were then tested for their ability to stimulate the production of Stx2 from a classic O157:H7 STEC strain. Based on the results of Maier et al. [4], we later also tested typical and atypical antipsychotic drugs. Fluoxetine and paroxetine were the most potent SSRIs as inducers of the SOS response and of Stx2 toxin production. A typical antipsychotic drug, trifluoperazine, also induced RecA expression and Stx toxin. 

SSRI antidepressants are heavily prescribed around the world. These drugs are resistant to breakdown in the environment and are not removed from sewage via treatment plants. Our results, together with recent findings that SSRIs are detectable in wastewater effluent and in natural surface waters in many countries, may provide a warning for the possible environmental effects of these drugs long after they have left the human body.

## 2. Results

As stated earlier, our strategy was to use the Miller assay with the *recA-lacZ* reporter strain for RecA induction as an initial screening assay and then to select drugs that activated RecA to test for their ability to induce production of Stx toxin from actual STEC strains. 

Figure 1A illustrates that fluoxetine, a serotonin selective reuptake inhibitor (SSRI), induced RecA expression in reporter strain JLM281. The results of testing with two other SSRIs is shown in Figure 1B, in which paroxetine also appeared to induce RecA expression, while duloxetine had a lesser effect on RecA. All three SSRI drugs were tested for their ability to trigger Stx production. 

Figure 1C illustrates that both fluoxetine and paroxetine were able to induce the production of Stx2 from STEC strain Popeye-1, the strain responsible for the 2006 spinach-associated outbreak in the United States. However, compared to paroxetine, fluoxetine was about 1.4-fold more potent in the induction of Stx2. Figure 1D displays a comparison of fluoxetine and duloxetine regarding their abilities to stimulate Stx2 production. Duloxetine appeared to induce Stx2 at lower concentrations (40 and 50 µg/mL), but Stx2 release then decreased at higher concentrations of duloxetine. The 60 and 70 µg/mL concentrations of duloxetine did strongly inhibit bacterial growth, but this growth inhibition was not accompanied by Stx2 production. Therefore, at concentrations of SSRIs in the 60 µg/mL range, the order of potency was fluoxetine > paroxetine > duloxetine.

Since Maier et al., reported that antipsychotic drugs were strong inhibitors of bacteria comprising the normal gut microbiome [4], we tested whether antipsychotic drugs also activated RecA. As depicted in Figure 2A, trifluoperazine, an older phenothiazine antipsychotic drug, activated RecA expression in a manner similar to that of fluoxetine (please compare Figure 2A with Figure 1A). Thioridazine induced RecA expression only weakly (Figure 2B), while loxapine and quetiapine did not appear to have any ability to trigger RecA expression (Figure 2C,D). This lack of RecA expression was interesting because both loxapine and quetiapine inhibited bacterial growth, as measured by culture turbidity (OD_600_; OD_600_ graphs not shown). 

As trifluoperazine demonstrated the ability to induce RecA expression, we also tested the ability of other antipsychotic drugs to trigger Stx2 production. We compared these effects with two SSRIs and two other classic inducers of the SOS response. Figure 2E illustrates that trifluoperazine induced Stx2 in amounts similar to fluoxetine. However, the effects of the psychoactive drugs were much less than that of zidovudine or ciprofloxacin. 

Several observations in Figure 1 and Figure 2 are worth emphasizing. First, for both the SSRIs and the antipsychotic drug trifluoperazine, the magnitude of the induction of Stx2 was far greater than that of the induction of the RecA reporter gene in the Miller assay using JLM281. This should not come as such a surprise, however, as our colleagues in the Koudelka laboratory have noted that the SOS response is much more inducible in STEC than in non-STEC *E. coli* strains, which they have called the “hair trigger” mechanism [8]. A second observation is that, within both the SSRI and antipsychotic drug classes, the agent with the greatest SOS and Stx-inducing activities was the drug containing a trifluoromethyl group (Figure 3). Indeed, for the SSRIs, the order of potency mentioned previously (fluoxetine > paroxetine > duloxetine) correlates roughly with the number of fluorine atoms within each molecule. This simplified and preliminary analysis needs to be expanded, though, with a larger number of drugs and more rigorous study of structure–activity relationships (SAR).

The induction of the SOS response is accompanied by marked elongation of bacterial cells, which is also called filamentation. This elongation is due to the arrest of cell division caused by inhibition of the FtsZ fission ring by SulA [9]. SulA is induced during the SOS response when the LexA repressor is cleaved [10,11]. We noted that bacterial elongation was very pronounced when the strains used in this study were exposed to classic SOS inducers (Figure 4, Panels A and B). Figure 4A exhibits strain JLM281 treated with aztreonam, and Figure 4B exhibits strain Popeye-1 treated with sublethal concentrations of ciprofloxacin. In contrast, fluoxetine at 60 µg/mL triggered little to no elongation in STEC Popeye-1 (Figure 4C). With fluoxetine at 70 µg/mL, an occasional elongated cell could be seen (red arrow), but the majority of the *E. coli* bacteria maintained normal sizes (Figure 4D), 2–4 µm; their shape, and their staining characteristics with acridine orange. This suggests that the SSRIs, although capable of inducing RecA expression and Stx toxin production (Figure 1), may not completely recapitulate the phenotype induced by the classic SOS inducers, at least at the concentrations tested in this study.

## 3. Discussion

The impetus for our study derives from previous studies demonstrating the inhibition of bacterial growth by psychoactive drugs [1,4]. Growth inhibition appears to be necessary but not sufficient to induce an SOS response in *E. coli* and other Gram-negative enteric bacteria. For example, antibiotics such as azithromycin, fosfomycin, and tigecycline are active in inhibiting *E. coli* growth but are inactive in inducing the SOS response or triggering Stx production from STEC [12]. Therefore, drugs that activate the SOS response and therefore also Stx production [7] appear to be a subset of agents within a larger set of drugs with antimicrobial effects.

The concentrations of SSRIs and antipsychotic drugs needed to induce the SOS response and Stx production, as shown in Figure 1 and Figure 2, are well above the achievable serum concentrations of these medications [13]. The concentrations of these orally administered drugs in the lumen of the GI tract, however, might be closer to those shown in Figure 1 and Figure 2, according to the calculations of Maier et al. [4] (please see Extended Data Figure 6). In addition, the SSRIs are increasingly administered as extended release or delayed release formulations, which have the effect of increasing the concentration of these drugs in the lower small intestine, where substantial reabsorption of intestinal fluid has already occurred and the lumenal volume of distribution is less. This is relevant because the distal small intestine has a more substantial microbiome than the upper GI tract, and therefore, the drugs have a greater potential to interfere with commensal microbes in that location [14]. In addition, STEC usually infects the colon, so any unabsorbed drug reaching the colon could interact with STEC pathogens present there, as well.

Kumar et al. recently reported that serotonin downregulated the virulence of STEC and that of a related mouse pathogen, *Citrobacter rodentium* [15]. Our results should not be construed as contradicting that work, as we studied psychoactive drugs rather than the effect of serotonin itself. A future direction of study would be to test if serotonin, alongside an SSRI, has additive or antagonistic effects compared to serotonin applied alone.

The SSRI class of antidepressants are very commonly prescribed drugs, along with the antihypertensive agents, and cholesterol-lowering statins [16]. Not only are these drugs heavily prescribed but also their use has been steadily increasing [17]. The use of SSRIs is so widespread that these drugs are routinely detectable in the effluent from modern sewage treatment plants [18,19] and, now, increasingly from surface waters, such as rivers and lakes. SSRIs are persistent in the environment and, similar to mercury and dichlorodiphenyl-trichlorethane (DDT), can concentrate in animals in the food chain, including fish [20]. These findings have raised concerns about the effects of these drugs in the environment long after they have left the bodies of those humans who consumed them originally [21]. Shiga toxin production can defend STEC bacteria from predation by aquatic protists, such as Tetrahymena [22]; thus, it may not be unreasonable to consider whether inducers of Shiga toxin, such as the psychoactive drugs studied here, could perturb aquatic food chains.

SOS-inducing drugs also trigger hypermutation in *E. coli* and other enteric bacteria, and this mutator phenotype promotes the emergence of new antibiotic resistance [5,23,24]. This could occur, for example, in wastewater treatment plants, where drugs such as SSRIs and antipsychotics co-mingle with large numbers of bacteria and antibiotics. The presence of SOS-inducing chemicals within this microbial soup could drive “experiments” in microbial evolution, some of which would carry unpleasant consequences for humans and the environment.

## 4. Materials and Methods

### 4.1. Bacterial Strains Used

The bacterial strains used are listed in Table 1. The bacteria were grown overnight in an LB broth at 37 °C with 300 rpm shaking and then subcultured into Luria–Bertani broth or DMEM liquid medium. A previously reported, “DMEM” refers to DMEM/F12 medium supplemented with 18 mM NaHCO_3_ and 25 mM HEPES, pH 7.4, but without serum or antibiotics. JLM281 was a kind gift received from Dr. Jay L. Mellies of Reed College, Portland, OR.

### 4.2. Materials

The SSRIs and antipsychotic drugs used were obtained from Cayman Chem., Ann Arbor, MI. The Luria–Bertani (LB) broth medium, ciprofloxacin, zidovudine, and mitomycin C used as positive controls for SOS induction, and X-gal and acridine orange were obtained from Sigma-Aldrich, now called Millipore Sigma. The O-nitrophenyl galactoside (ONPG) substrate used in the Miller assay was obtained from Rockland Immunochemicals, Limerick, PA. DMEM/F12 powder was obtained from the Gibco division of Thermo-Fisher, Grand Island, NY.

### 4.3. Miller Assay for Expression of ß-Galactosidase in Bacterial Reporter Strains

Strain JLM281, the reporter strain containing the *recA-lacZ* construct, was used to measure RecA expression in response to inducing antibiotics, SSRIs, and antipsychotic drugs. Expression of RecA is an early and quantifiable marker of the SOS response in *E. coli*. JLM281 was subcultured at a 1:100 dilution overnight into DMEM medium for 1 to 1.5 h without that addition of any test substances. Putative SOS inducers, such as SSRIs, were then added, and incubation was continued in a shaker platform at 37 °C in a 96-well plate, as described [5]. RecA expression was measured in quadruplicate wells at 4 h after initiation of the culture, unless stated otherwise.

We used a version of the Miller assay adapted to 96-well plates for higher throughput [29]. However, we used 0.1% hexadecyltrimethylammonium bromide (HTA-Br) detergent alone, without chloroform or sodium dodecyl sulfate (SDS), to permeabilize the bacteria.

The SDS is omitted to avoid formation of foamy bubbles, which interfere with the measurement of the OD_600_ in the 96-well plate reader. The buffers used are described in an Open WetWare website at http://openwetware.org/wiki/Beta-Galactosidase_Assay_%28A_better_Miller%29 (accessed on 29 April 2021).

We used the same conditions for the Miller assay as reported previously [5] and, again, omitted the addition of the Na_2_CO_3_ STOP buffer.

### 4.4. Induction of Stx Expression and Enzyme Immunoassay (EIA) for Shiga Toxin Protein

As previously described, strain Popeye-1 was allowed to grow in DMEM for 1 h before the addition of SOS-inducing drugs. For strain Popeye-1, we previously determined that a 5-hour period of growth in DMEM was optimal to allow time for the bacteria to produce Stx and for the Stx to be released from the bacterial cell by phage-induced lysis. Cultures were collected at the 5 h mark and centrifuged at 2500 *g* for 15 min to pellet the majority of bacteria, after which the supernatants were subjected to sterile filtration using syringe-tip filters.

### 4.5. Stx EIA

The Shiga toxin protein was measured by EIA using the Premier EHEC EIA kit obtained from Meridian Bioscience, Cincinnati, OH, which measures Stx1 and Stx2. However, the STEC strain we used, Popeye-1, only produces Stx2. The optimal dilutions for EIA detection had to be determined by trial and error, as described in [26]. To quantitate Stx2, we performed standard curves using the Stx2 toxoid that we received as a gift from Dr. Allison Weiss of the University of Cincinnati, OH, as previously described [27]. Stx EIAs were run in duplicate or triplicate.

### 4.6. Acridine Orange Staining

The bacteria were stained using 0.2% acridine orange dye in 50% ethanol for 15 min in a 37 °C bead bath. The tubes were centrifuged in the Eppendorf centrifuge for 2.5 min at room temperature, and the supernatant was decanted onto a paper towel. The pellet was resuspended in 350 µL of water by vortexing, and then, the centrifugation step was repeated. The bacteria were again decanted and then resuspended in the residual liquid, usually about 40 µL. The resuspended material was spread thinly on glass microscope slides, allowed to dry on a warming block, and photographed at 1000× under oil in the fluorescence microscope.

### 4.7. Data Analysis and Graphing

The graphs were generated using GraphPad Prism, version 9.1. Statistical analysis was conducted by *t*-test or ANOVA, and the error bars shown are standard deviations. Where error bars do not appear obvious, it is because the error bars were smaller than the symbol(s) used.

## Figures and Tables

**Figure 1 toxins-13-00437-f001:**
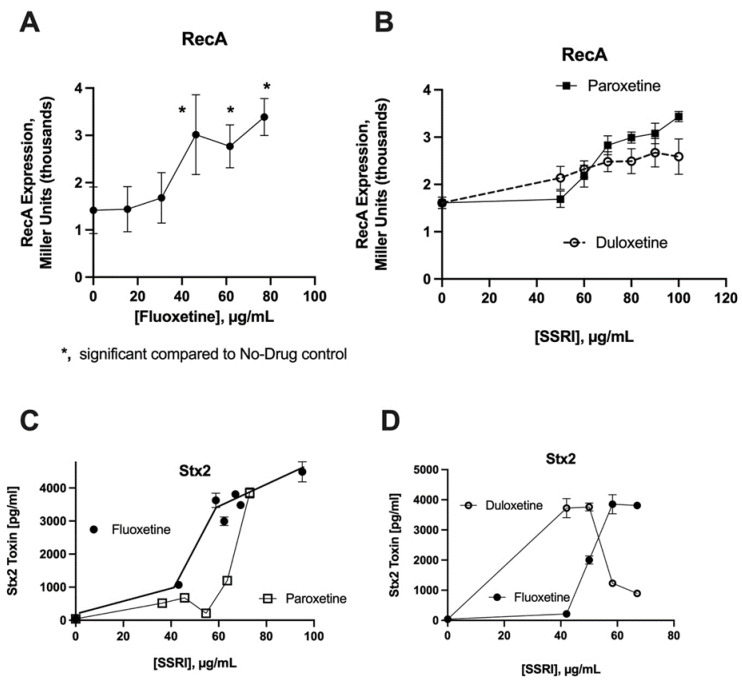
Effect of SSRIs on RecA expression and Stx2 release. Panels (**A**,**B**): RecA expression was measured using the JLM281 reporter strain and the Miller assay, as described in the Materials and Methods section. Panels (**C**,**D**): Stx2 production and release into the supernatant medium was measured at 5 h in response to three SSRIs, using STEC strain Popeye-1.

**Figure 2 toxins-13-00437-f002:**
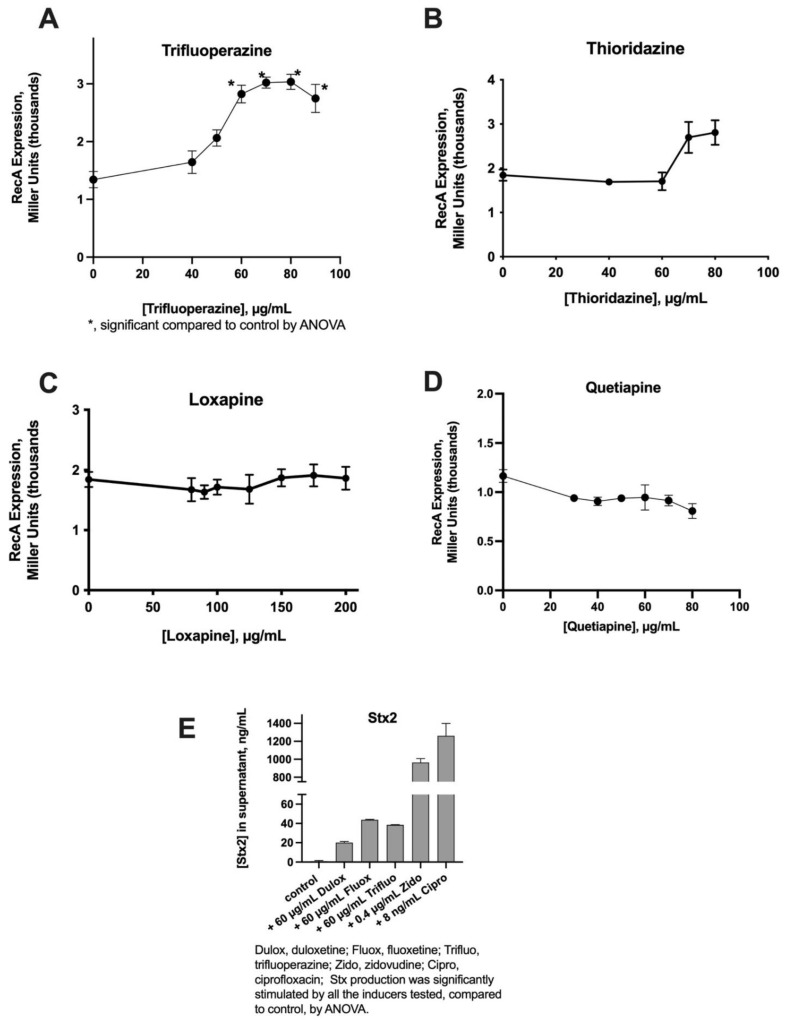
Effect of antipsychotic drugs on RecA expression and on Stx production. Panels (**A**–**D**): RecA expression using the Miller assay. Panel (**E**): comparison of Stx production triggered by two SSRIs, by the antipsychotic drug trifluoperazine, by zidovudine, and by ciprofloxacin. Stx2 release from Popeye-1 was measured at 5 h.

**Figure 3 toxins-13-00437-f003:**
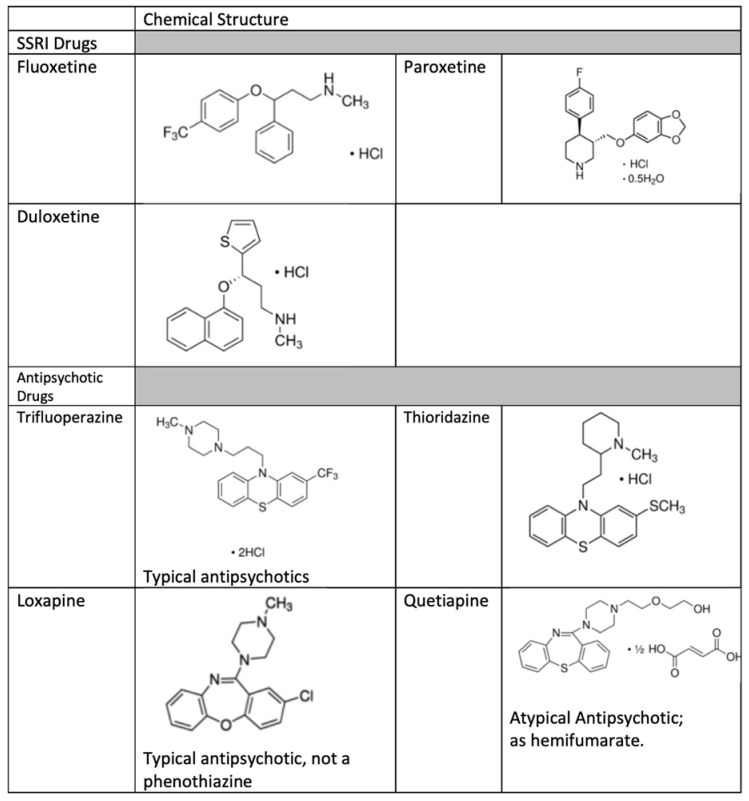
Structures of several SSRI antidepressant and antipsychotic drugs.

**Figure 4 toxins-13-00437-f004:**
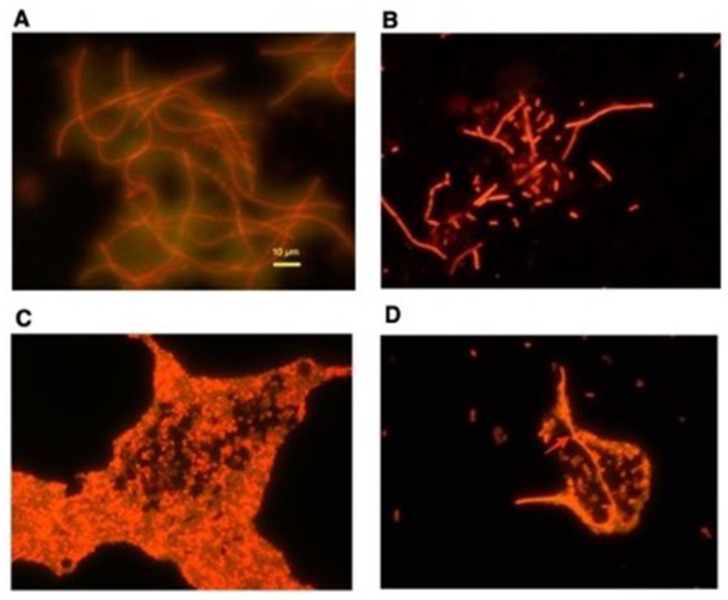
Elongation response of *E. coli* bacteria to aztreonam, ciprofloxacin, and fluoxetine. Panel (**A**): *E. coli* JLM281 was exposed to 0.75 µg/mL aztreonam, beginning 1 h after subculture. Bacteria were collected at 4 h, stained with 0.2% acridine orange in 50% ethanol, then washed, allowed to dry on a glass microscope slide, and photographed at 1000× magnification. Size bar indicates 10 µm. Panel (**B**): *E. coli* Popeye-1 was exposed to 12 ng/mL ciprofloxacin, added after a 1 h delay, and was collected at 5 h for acridine orange staining. Panels (**C**,**D**): strain Popeye-1 was exposed to either 60 µg/mL fluoxetine (Panel **C**) or 70 µg/mL fluoxetine (Panel **D**) and then collected, stained, and photographed. The red arrow in Panel (**D**) indicates an elongated *E. coli* bacterial cell. The size bar in Figure 4A applies to the other panels of Figure 4 as well.

**Table 1 toxins-13-00437-t001:** Description of bacterial strains used.

*E. coli* Strain	Description	Comments	Reference
JLM281	*recA-lacZ* reporter strain	Measures RecA expression via ß-galactosidase	[5,25]
Popeye-1	STEC O157:H7, from 2006 spinach-associated outbreak in U.S.	TW14359; Stx2, Stx2c producer	[26,27,28]

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
