# Peer review of "Psychoactive Drugs Induce the SOS Response and Shiga Toxin Production in Escherichia coli"

_toxins, 2021, doi:10.3390/toxins13070437_

Round 1
Reviewer 1 Report
Interesting paper on the effect of Psychoactive drugs of the SOS response and stx production in E. coli. The paper is well written and I have only some minor comments:
- Page 1, line 15: should be a full stop instead of comma between "humans" and "In".
- Page 2, line 78: "at" instead of "as" in "... to induce Stx2 as lower ..."
- Figure 1: It would be easier to read if the name of the drug was listes similarly om all four panels. The drug name in fig 1 A is placed below the figure, while in the three other panels the drug names are listed in the panel.
- Figure 2 legend: Stx2 release from Popeye-1 was again measured at 5 h. Why "again": was it not measured just one time?
- Page 5, line 11: here is referred to "Table 1": the content of table 1 looks like a figure to me. I think it is not bnecessary to presnet figures of the molecular structure of the drugs as a figure in the paper, and not as part of the results. This would probably fit better as part of the discussion section, or even could be mentioned in the introduction.
- Figure 3: I cannot find a description of this in the methods section, only in the figure legend. Methods used to produce results should be presented in the methods section.
- Page 9, line 22: Is this sentence meant to be a heading, if not it is not a complete sentence.
- Methods section: generally very short description of methods used. Could preferably have been more detailed. Eg. I cannot find any description in the method section or in the figure legends how many replicates that were done in the analyses of recA expression, and in the analysis of stx2 expression.
Author Response
Reviewer 1
Interesting paper on the effect of Psychoactive drugs of the SOS response and stx production in E. coli. The paper is well written and I have only some minor comments:
- Page 1, line 15: should be a full stop instead of comma between "humans" and "In".
This has been fixed.
- Page 2, line 78: "at" instead of "as" in "... to induce Stx2 as lower ..."
This has been fixed.
- Figure 1: It would be easier to read if the name of the drug was listes similarly om all four panels. The drug name in fig 1 A is placed below the figure, while in the three other panels the drug names are listed in the panel.
This was because Panels B, C, and D, compared 2 drugs per figure, while Panel A had only 1 drug. “Fluoxetine” has been added to Fig. 1A to make it consistent.
- Figure 2 legend: Stx2 release from Popeye-1 was again measured at 5 h. Why "again": was it not measured just one time? -ïƒ “Again” has been deleted.
- Page 5, line 11: here is referred to "Table 1": the content of table 1 looks like a figure to me. I think it is not bnecessary to presnet figures of the molecular structure of the drugs as a figure in the paper, and not as part of the results. This would probably fit better as part of the discussion section, or even could be mentioned in the introduction. ïƒ Table 1 has been converted to a Figure at the request of Reviewer 1, the new Figure 3.
- Figure 3: I cannot find a description of this in the methods section, only in the figure legend. Methods used to produce results should be presented in the methods section.
ïƒ The staining method has been added to the Method Section.
- Page 9, line 22: Is this sentence meant to be a heading, if not it is not a complete sentence.
ïƒ Yes, of course, this was a Heading. It has been converted to Italics and given a number.
- Methods section: generally very short description of methods used. Could preferably have been more detailed. Eg. I cannot find any description in the method section or in the figure legends how many replicates that were done in the analyses of recA expression, and in the analysis of stx2 expression.
ïƒ The Methods are not elaborate because we have published the detailed descriptions of the Methods many times before. We did add mention of the number of replicates in the Miller assay (4, quadruplicates) and in the Stx EIA (2 or 3)
Reviewer 2 Report
- Modest English language editing is required.
- A control has to be shown in the figures, the number of replicates must be stated and where significant differences were encountered (as in Figure 1 and 2A). The significant differences should be referred to in the text.
Author Response
Reviewer 2.
- Modest English language editing is required. - √ Okay
- A control has to be shown in the figures, the number of replicates must be stated and where significant differences were encountered (as in Figure 1 and 2A). The significant differences should be referred to in the text.
All of the Figures have controls. For Figs. 1 and 2, for example, the condition receiving Zero Drug (on the y-axis) is the control. We have added a comment about the statistical significance in Fig. 2E (Bar Graph).
Reviewer 3 Report
Toxins, entitled „Psychoactive Drugs Induce the SOS Response and Shiga Toxin Production in Escherichia coli”.
The authors analysed the induction and expression of RecA corresponding SOS response in E. coli JLM281 as recA-lacZ reporter strain and of Shiga toxins corresponding Stx proteins in the STEC strain Popeye-1. The induction and expression of the Shiga toxins Stx2 and 2c expressed by the STEC O157:H7 strain was analysed by application of psychoactive drugs. Several psychoactive drugs indicate strong antimicrobial properties and inhibition of bacterial growth. Additional they induce bacterial SOS responses. On the other hand, others are inactive in inducing SOS response and Stx production but inhibit bacterial growth. In this study the expression of the Shiga toxins in strain Popeye-1 was analysed in comparison with an SOS response in JLM281.
It is an interesting study regarding the expression of RecA and Stx induced by psychotropic drugs. The authors showed an effect on SOS response and Stx expression. To demonstrate this effect, two E. coli strains were studied regarding different parameters. Unfortunately, there was no common characterisation based on the two strains, thus the results show small generality. I see this point of moderate comparability as critical in the study. It would be beneficial to choose biochemical characteristics that can be studied in the two strains or more comparable STEC strains, because the study focused on Stx expression caused by drugs. This is because it has been shown time and again that biochemical characteristics can differ, especially with respect to induction and expression.
The results in the manuscript are clearly presented. However, Stx expression results shown in figure 1 should be explained in more detail (see below).
Here are some detail remarks:
Material and Methods:
To 4.1. The comments to the strains should not be included in the table but in the text.
JLM281 is the strain and the gift that should be mentioned in 4.1, not in 4.2.
What is LB: Luria broth or lysogenic broth? Please add the reference company and the reference state.
In total: please add the companies from which you purchased the chemicals
Line 192: 37°C
Line 72: the word “strains” in this context does not refer to bacterial strains and therefore confuses here. “Substances” seems to be a more appropriate word.
Line 76: reference of the spinach-associated outbreak in the USA is missing
Line 84: Number of reference should be included
Line 87: e.g. “did activate”: use “activated” There are several such descriptions. It is recommended to have the manuscript revised by a native speaker.
Figure 1C and 1D: The Stx2 toxin expression is analysed with Paroxetine and Fluorxetine in C and Duloxetine and Fluoxetine in D. Thus, the Stx toxin expression curves with Fluoxetine are given in both figures. However, both curves differ. Is it due to different experiments or are both substances tested in one approach? These different representations are not comprehensible in this way and require clarification of the experimental procedure. If there are different approaches, there should be an explanation why at about 45 µg Fluoxetine in C the toxin production increases up to 1000 pg/ml, while in D it is about only 100 pg/ml.
Figure 1C: The Stx2 expression line increased but goes down at app. 55 µg/ml Paroxetine and strongly increases with > 60 µg/ml. Do you have an explanation for this phenomenon? There are no error bars or standard deviation. Was the experiment performed only once?
Figure 1 and 2: labelling of the y-axis: RecA expression (next line) Miller units (x 103); significance should be written in the legend text with indication of the statistical calculation.
Figure 1 e.g. B, valid for all figures: Designations and symbols should be included in the legend text and not in the illustration.
Figure 2E: Please give an explanation why you have chosen these exact concentrations.
The legend texts of figures 1 and 2 and 3 should be described in more “Material & Methods” details
Table 1: Structures of some...
Author Response
Reviewer 3.
The authors analysed the induction and expression of RecA corresponding SOS response in E. coli JLM281 as recA-lacZ reporter strain and of Shiga toxins corresponding Stx proteins in the STEC strain Popeye-1. The induction and expression of the Shiga toxins Stx2 and 2c expressed by the STEC O157:H7 strain was analysed by application of psychoactive drugs. Several psychoactive drugs indicate strong antimicrobial properties and inhibition of bacterial growth. Additional they induce bacterial SOS responses. On the other hand, others are inactive in inducing SOS response and Stx production but inhibit bacterial growth. In this study the expression of the Shiga toxins in strain Popeye-1 was analysed in comparison with an SOS response in JLM281.
It is an interesting study regarding the expression of RecA and Stx induced by psychotropic drugs. The authors showed an effect on SOS response and Stx expression. To demonstrate this effect, two E. coli strains were studied regarding different parameters. Unfortunately, there was no common characterisation based on the two strains, thus the results show small generality. I see this point of moderate comparability as critical in the study. It would be beneficial to choose biochemical characteristics that can be studied in the two strains or more comparable STEC strains, because the study focused on Stx expression caused by drugs. This is because it has been shown time and again that biochemical characteristics can differ, especially with respect to induction and expression.
ïƒ It was difficult to figure out how to respond to this critique. JLM281 and Popeye-1 are not closely related to one another within the species E. coli. While Reviewer 3 found this to be a limitation, I think it is a strength because it increases the generalizability of our findings since two strains show consistency in their SOS response to the various inducers. With greater funding, it would be interesting to analyze RNA expression, for example by RNA Seq, from a selection of diverse E. coli strains and to compare the RNA responses to “Classical” SOS inducers such as ciprofloxacin, zidovudine, and mitomycin C, on the one hand, with the “Atypical” inducers such as fluoxetine and trifluoperazine, on the other.
The results in the manuscript are clearly presented. However, Stx expression results shown in figure 1 should be explained in more detail (see below).
Here are some detail remarks:
Material and Methods:
To 4.1. The comments to the strains should not be included in the table but in the text.
ïƒ A description of the bacterial strains IS allowed in the table. Please see the Instructions to Authors, especially the “Free Format Submission” section, in which authors are granted more leeway in formatting that in years and decades past. https://www.mdpi.com/journal/toxins/instructions
JLM281 is the strain and the gift that should be mentioned in 4.1, not in 4.2.
ïƒ This was fixed.
What is LB: Luria broth or lysogenic broth? Please add the reference company and the reference state.
ïƒ Luria-Bertani broth; this has been fixed.
In total: please add the companies from which you purchased the chemicals.
ïƒ The suppliers have been included in the Materials and Methods.
Line 192: 37°C ïƒ C for Celsius has been added
Line 72: the word “strains” in this context does not refer to bacterial strains and therefore confuses here. “Substances” seems to be a more appropriate word.
ïƒ This was fixed and the word “drugs” was substituted.
Line 76: reference of the spinach-associated outbreak in the USA is missing
ïƒ We inserted a reference to the 2006 Spinach-Associated outbreak.
Line 84: Number of reference should be included. ïƒ Reference to Maier et al. inserted.
Line 87: e.g. “did activate”: use “activated” There are several such descriptions. It is recommended to have the manuscript revised by a native speaker.
We used “did activate” for emphasis. All of the authors are native English speakers. My mother was a journalist and a newspaper editor (and a strict one). My aunt was an English teacher. I attended Cornell University where we were steeped in the precepts from the “Elements of Style” by Strunk and White. Nevertheless, we changed the wording to “activated” at the request of Reviewer 3.
Figure 1C and 1D: The Stx2 toxin expression is analysed with Paroxetine and Fluorxetine in C and Duloxetine and Fluoxetine in D. Thus, the Stx toxin expression curves with Fluoxetine are given in both figures. However, both curves differ. Is it due to different experiments or are both substances tested in one approach? These different representations are not comprehensible in this way and require clarification of the experimental procedure. If there are different approaches, there should be an explanation why at about 45 µg Fluoxetine in C the toxin production increases up to 1000 pg/ml, while in D it is about only 100 pg/ml.
The figures show the experiment-to-experiment variation in the amount of Stx2 released. At the 5 hour time point, Stx release is going up very quickly, so small variations in how quickly the specimens are collected, centrifuged, and processed can make a difference in the data. There are a large number of specimens that need to be processed in these experiments.
Figure 1C: The Stx2 expression line increased but goes down at app. 55 µg/ml Paroxetine and strongly increases with > 60 µg/ml. Do you have an explanation for this phenomenon? There are no error bars or standard deviation. Was the experiment performed only once?
ïƒ The small dip in the Stx released in the paroxetine curve in Fig. 1C corresponded to concentration that caused a drop in growth, so perhaps there were fewer bacteria and lower Stx. At higher concentrations, 70 to 80 µg/mL paroxetine, the SOS-mediated induction of Stx begins to take effect and Stx production then goes up quickly. “no error bars or standard deviation” Not True. In the few cases where error bars are not readily visible, it is because the error bar was smaller than the symbol used.
Figure 1 and 2: labelling of the y-axis: RecA expression (next line) Miller units (x 103); significance should be written in the legend text with indication of the statistical calculation.
Once again, see the Instructions to Authors, especially the “Free Format Submission” section, in which authors are granted more leeway in formatting that in years and decades past. https://www.mdpi.com/journal/toxins/instructions
Figure 1 e.g. B, valid for all figures: Designations and symbols should be included in the legend text and not in the illustration.
ïƒ Not true. Please see Instructions to Authors.
Figure 2E: Please give an explanation why you have chosen these exact concentrations.
ïƒ The rationale for using these concentrations is stated in the Results, where we stated:
“…The 60 and 70 µg/mL concentrations of duloxetine did strongly inhibit bacterial growth, but this growth inhibition was not accompanied by Stx2 production. Therefore, at concentrations of SSRIs in the 60 µg/mL range, the order of potency was fluoxetine > paroxetine > duloxetine.”
The legend texts of figures 1 and 2 and 3 should be described in more “Material & Methods” details.
The Materials and Methods section was expanded as mentioned above.
Table 1: Structures of some...
ïƒ Table 1 was converted to Fig. 3 as requested by Reviewer 1 abov

Round 2
Reviewer 3 Report
Thank you very much for the critical revisions and additional explanations. Now, it sounds clear and very good.